# Spatially resolved analysis of *Pseudomonas aeruginosa* biofilm proteomes measured by laser ablation sample transfer

Aruni Chathurya Pulukkody[1], Yeni P. Yung[1], Fabrizio Donnarumma[2], Kermit K. Murray[2], Ross P. Carlson[3], Luke Hanley[1] *

**1** Department of Chemistry, University of Illinois at Chicago, Chicago, Illinois, United States of America,
**2** Department of Chemistry, Louisiana State University, Baton Rouge, Louisiana, United States of America,
**3** Department of Chemical and Biological Engineering, Center for Biofilm Engineering, Montana State University, Bozeman, Montana, United States of America

* lhanley@uic.edu

**Data Availability Statement:** The proteomic data that support the findings of this study are available on the Mass Spectrometry Interactive Virtual

## Abstract

Heterogeneity in the distribution of nutrients and oxygen gradients during biofilm growth gives rise to changes in phenotype. There has been long term interest in identifying spatial differences during biofilm development including clues that identify chemical heterogeneity. Laser ablation sample transfer (LAST) allows site-specific sampling combined with label free proteomics to distinguish radially and axially resolved proteomes for *Pseudomonas aeruginosa* biofilms. Specifically, differential protein abundances on oxic vs. anoxic regions of a biofilm were observed by combining LAST with bottom up proteomics. This study reveals a more active metabolism in the anoxic region of the biofilm with respect to the oxic region for this clinical strain of *P. aeruginosa*, despite this organism being considered an aerobe by nature. Protein abundance data related to cellular acclimations to chemical gradients include identification of glucose catabolizing proteins, high abundance of proteins from arginine and polyamine metabolism, and proteins that could also support virulence and environmental stress mediation in the anoxic region. Finally, the LAST methodology requires only a few mm$^2$ of biofilm area to identify hundreds of proteins.

## Introduction

*Pseudomonas aeruginosa* is a metabolically versatile bacteria that synchronizes expression of genes in a controlled framework such that survival in different environments is matched to manage efficiency and fitness during its growth stages [1]. *P. aeruginosa* can generate bacterial virulence factors that amplify antibiotic resistance, making it an opportunistic pathogen that takes advantage of immunocompromised hosts. Thus, *P. aeruginosa* is prominent within infections that occur in patients with cystic fibrosis, diabetic foot ulcers, burn wounds, and implanted bio-materials [2]. Because such nosocomial infections can transition quickly from acute to chronic stages, they can be difficult to treat and even life threatening [3]. The severity of such infections arises in part from antibiotic resistance which causes morbidity and mortality in patients, with over two million cases within the U.S. population annually [4, 5].

Environment (MassIVE) repository at ftp://massive.ucsd.edu/MSV000086320/.

**Funding:** LH & RPC were awarded National Institute of Biomedical Imaging and Bioengineering grant 1 U01 EB019416, nih.gov None of the authors received National Center for Research Resources grant 1 S10 RR025653-01A1 (this was an instrumentation grant awarded to a different investigator at the University of Illinois at Chicago). The funders had no role in study design, data collection and analysis, decision to publish, or preparation of the manuscript.

**Competing interests:** The authors have declared that no competing interests exist.

The sessile aggregates or biofilms of *P. aeruginosa* have been widely studied and chemical and physical heterogeneity along radial and axial (vertical) axes are common [6]. Axial distribution of oxygen in *P. aeruginosa* biofilms has been measured using microelectrodes [7, 8], while metabolic activity has been measured using reporter proteins [6, 9, 10] and transcriptomic analyses [6, 9]. Proteomic analyses of whole biofilms have also been performed previously [8, 11–13]. However, there have been relatively few studies focused on spatially resolved *P. aeruginosa* biofilm proteomics [6, 14, 15].

Mass spectrometry (MS) imaging studies the spatial distribution of multiple analytes ranging from small molecules to lipids, peptides and proteins by creating a two-dimensional heat map of the abundance and distribution of representative ions [15]. Matrix assisted laser desorption ionization (MALDI) is the most popular probe for MS imaging [16]. MALDI usually employs nanosecond (ns), ultraviolet laser pulses to desorb biomolecules from a chemical matrix-coated sample [17]. Improvements in spatial resolution and sensitivity in MALDI-MS imaging have been achieved using high mass resolution/accuracy MS instrumentation, methodologies to spray thinner and more uniform matrix coatings, new matrices, and better laser spot focusing [15, 17–19]. Another popular method in MS imaging crosses the ablation plume formed by a mid-infrared ns laser pulse with the output from an electrospray ionization source (sometimes referred to as laser ablation electrospray ionization or LAESI) [20]. Laser wavelengths near ~3 μm are resonant with the OH stretch vibrations of water, facilitating explosive plume-like ablation from biofilms and other hydrated biological samples [20]. MALDI, LAESI, and other popular methods of MS imaging like desorption electrospray ionization (DESI) [21, 22] typically lack the chromatographic separation that is so effective for MS analysis of bacterial biofilms, animal tissue, and other intact biological samples.

Localized sampling can be accomplished in several ways beyond the use of focused probes. Manual dissection [23–25] can remove a portion of the surface from the bulk material, while laser capture microdissection [26–28] allows more precise localized sampling. Liquid microjunctions [29] have also been used to achieve extraction of specific analytes from a biological surface through tuning the composition of the extraction mixture. However, it is either difficult or impossible to perform a layered extraction or measure a depth profile using these methods.

Laser ablation sample transfer (LAST) is utilized here for sampling, as shown in Fig 1. LAST utilizes a mid-infrared ns-pulsed laser to ablate hydrated biological samples [30, 31], like LAESI. Murray and co-workers have used LAST for analysis of a diversity of samples including mouse brain, mouse lung tissue, peptide and protein standards, fingerprints, ink, and both enzymes and lipids on bacterial colonies of *Escherichia coli* and *Bacillus cereus* grown on liquid media [30–34]. The LAST method can be used as a field sampling or offline technique [35], where collected material can be stored for subsequent processing and analysis. LAST also permits spatially resolved sampling and can be sequentially combined with liquid chromatographic (LC) separation and tandem mass spectrometry (MS/MS) for bottom-up proteomic analysis of complex biological samples.

Selection of the laser fluence and irradiation geometry enables different ablation modalities. Higher laser fluence irradiation in reflection geometry (front side) can ablate the entire irradiated biofilm volume. Lower laser fluence irradiation in transmission geometry (back side of the glass slide, through translucent biofilm) induces laser spallation [36, 37] that leads to ablation of only the outmost layer of the biofilm, as shown in Fig 2. Spallation is thought to proceed via a pressure wave propagating from the point of irradiation along the longitudinal axis of the biofilm [36, 37].

LAST is demonstrated here to be effective for region-specific proteomic analysis of bacterial biofilms by coupling it with bottom-up proteomics via LC and high-resolution Orbitrap MS/MS. The method is found here to be particularly sensitive in that it requires only a few mm$^2$

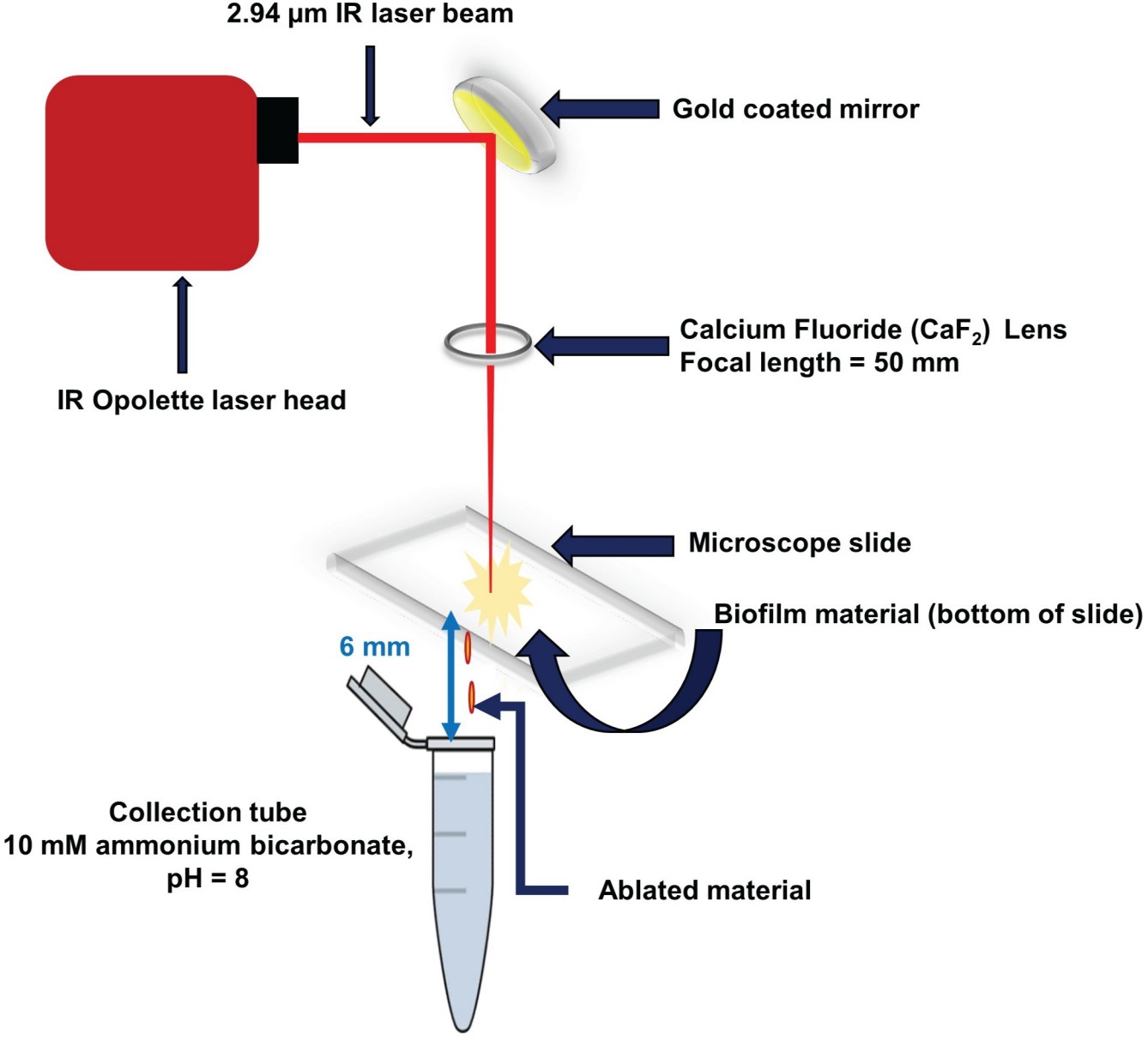

**Fig 1. Schematic of laser ablation sample transfer (LAST) apparatus.** A mid-IR laser beam is directed in transmission geometry through a glass slide to irradiate a biofilm. The spallated material is collected on a microcentrifuge tube for further processing.

area of a biofilm to identify hundreds of proteins. This work combines current advances in bottom-up proteomics to spatially resolve bacterial proteomes within *P. aeruginosa* biofilms, further revealing the remarkable molecular machinery involved in their acclimation to environmental conditions.

## Materials and methods

### Laser ablation sample transfer

The laser ablation sample transfer setup was constructed in-house based upon a published design of Murray and Donnarumma [33] and is depicted in Fig 1. Mid-infrared laser pulses

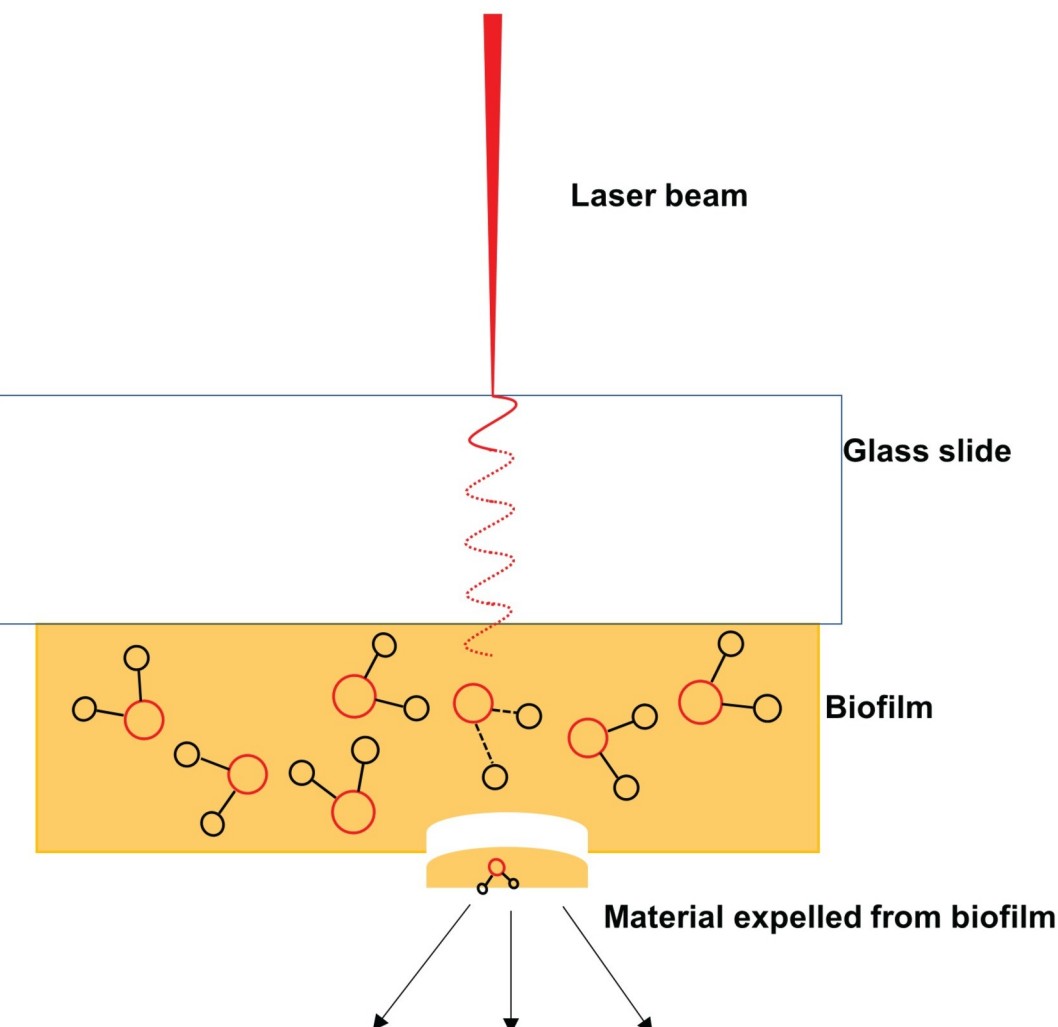

**Fig 2. Ejection of biofilm via spallation.** Spallation is thought to proceed via pressure wave propagation from the point of irradiation along the longitudinal axis of the back irradiated biofilm.

(2.94 μm, ~7 ns, 20 Hz) from a Nd:YAG-pumped optical parametric oscillator (IR Opolette, Opotek, Carlsbad, CA) were transmitted through an iris onto a 45˚ angle gold mirror. The reflected laser beam was focused normal to the sample surface through a 50 mm CaF$_2$ lens in transmission geometry onto biofilm samples adhered to glass microscope slides. Inverted sample slides were placed on a xyz translation stage operated using 25 mm motorized linear actuators (Model 462, CMA-25CCCL, Newport, Irvine, CA, USA), controlled by previously described software [38]. Biofilm samples were ablated with the laser irradiation perpendicular to the sample surface while the sample stage moved at a speed of 0.2 mm/s. The spot size of the focused beam was 260 μm with a laser fluence of 4.6 ± 2.1 J/cm$^2$. The laser spot size was determined by ablation of agar gels, followed by examination of the gels using a digital camera (DFC 495, Leica Microsystems Inc., Buffalo Grove, IL), where ~50 μm thick agar gels were prepared from 1.5 g of agar powder dissolved in 5.0 mL of water, coated on glass microscope slides and air-dried. Ablation followed the perimeter of 4 mm square of a biofilm, requiring a total of ~1500 laser shots. A microcentrifuge tube with buffered solution (10 mM NH$_4$HCO$_3$, pH 8.0) was used to collect the ablated material.

## Growth and ablation of biofilms

Chronic wound clinical isolate 215 strain of *Pseudomonas aeruginosa* was chosen for these studies as it was part of a polymicrobial wound biofilm model [39]. A glycerol stock of *P. aeruginosa* was used to inoculate a CSP-agar (chemically defined nutrient media for bacterial growth) streak plate, as described previously [8]. The streak plate inoculum was incubated for 16 hours at 37°C, then used to establish fresh colonies of *P. aeruginosa* in planktonic culture (10 mL of CSP liquid media). This planktonic culture was grown until it attained an optical density of 0.1 at 600 nm absorbance ($OD_{600}$).

Planktonic cultures of *P. aeruginosa* at $OD_{600}$ = 0.1 (20 μL) were inoculated onto sterile polycarbonate membranes (25 mm diam, 0.2 μm pore size from GE Life Sciences, Marlborough, MA), then were grown for three days on chemically-defined (CSP) medium introduced into solid agar. Biofilms supported on membranes were transferred onto fresh media plates every 24 hours [8]. These polycarbonate-grown biofilms were used to develop and evaluate the analytical methodology and to measure the radially resolved proteome.

Biofilms were also grown on transwell cell culture plate membrane inserts (0.4 μm pore size, Corning Inc. Tewksbury, PA) that were placed in six well plates with 2.5 mL of CSP media. The transwell membranes were inoculated with 100 μL of diluted *P. aeruginosa* culture (from $OD_{600}$ = 0.1 aliquots of planktonic culture which were first diluted to $OD_{600} \approx 0.03$) and grown for three days with agitation. The transwell membranes were transferred every 12 hours to fresh CSP media in new cell culture plates. A scalpel blade was used to separate the transwell membranes from the plastic insert that was then inverted onto a microscope slide. The glass slide was then placed in a secondary container and stored in dry ice for 15 min, the membrane was peeled off while still frozen to expose its anoxic side (see S1 Fig), and finally thawed prior to laser irradiation for examination of the axially resolved proteome. A similar process was used to expose the oxic surface via a second transwell membrane, as shown in Fig 3.

## Proteomic analyses

The basic LC-MS/MS analysis strategy as well as the qualitative and quantitative proteomics were adapted from a method that was previously described in detail [8] and is only summarized here. Laser ablated material from the polycarbonate- or transwell-membrane biofilms was captured in a buffered solution, then concentrated by vacuum centrifugation (Labconco Corp., Kansas City, MO) at 30°C and 1725 rpm. The residues were reconstituted in 20 μL of Laemmli buffer with 5% β-mercaptoethanol to reduce disulfide bonds, then heated to 90°C for complete denaturation of proteins for sodium dodecyl sulphate—polyacrylamide gel electrophoresis (SDS-PAGE) analysis.

Each lane in the SDS-PAGE gel was excised and processed with an in-gel digestion protocol [40]. The processing of entire lanes was done to maintain consistency with the SDS-PAGE results. The resultant tryptic digested peptides were reconstituted in 10 μL of 5% acetonitrile, 0.1% formic acid buffer for LC-MS/MS analysis. Difficulties arose in quantifying the relatively low levels of protein by a Bradford assay [41, 42]. Peptide concentrations for LC-MS/MS injection from the transwell membranes were instead measured using 1 μL aliquots of each sample on a micro-volume UV-visible spectrophotometer (Nano Drop One, Thermo Fisher Scientific, Wilmington, DE) [43], where the absorbance ratios of 280 nm to 205 nm (due to amide bonds of proteins) were used to determine the volume of injection corresponding to an equivalent amount of protein for normalization across biological replicates. This strategy reduced interference in UV absorbance from detergents and also reduced bias arising from consideration of only aromatic peptides.

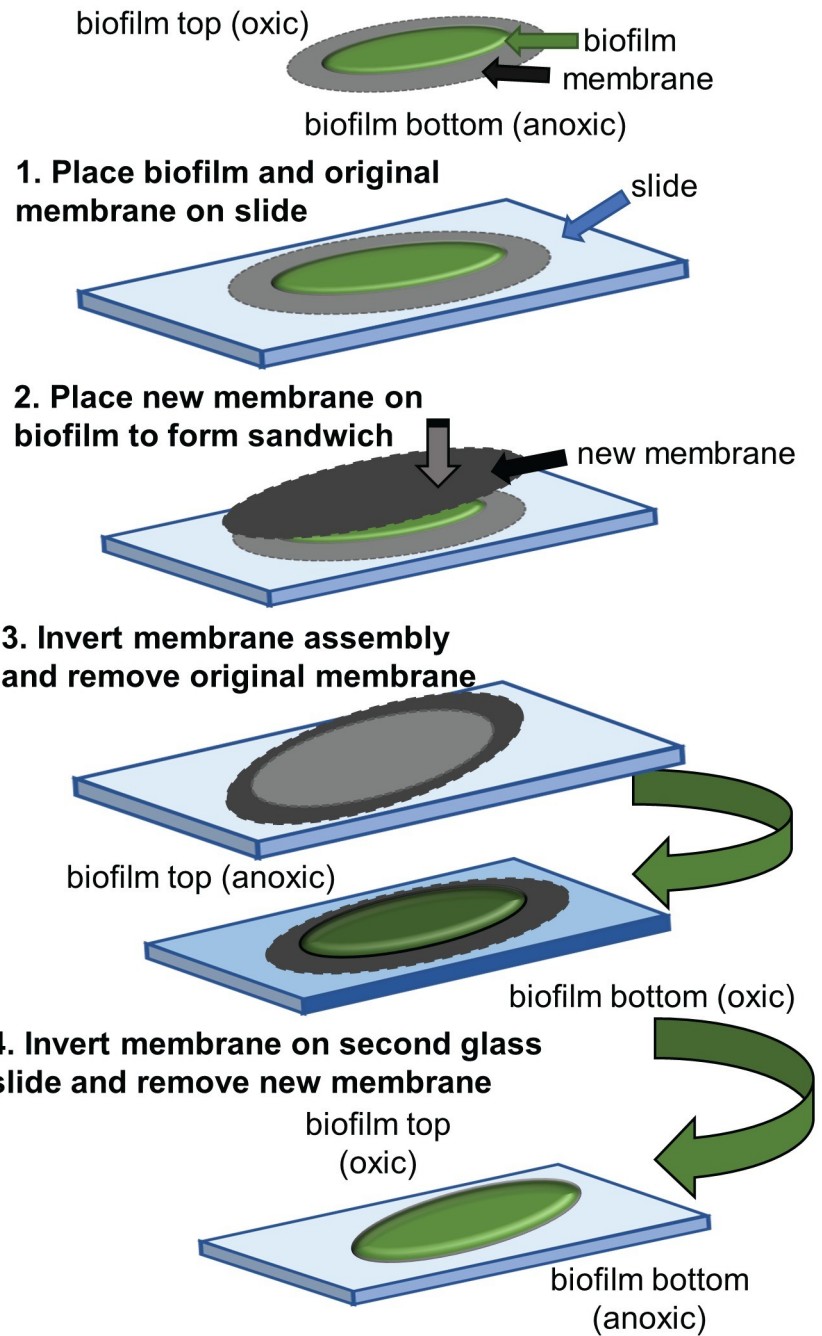

**Fig 3. Technique for exposing oxic side of biofilm.** The exposed, oxic side of a biofilm grown on its original membrane was placed on a glass slide, then covered with a new membrane. The membrane-biofilm sandwich was inverted to peel off the old membrane. The exposed anoxic side on the new membrane was then inverted onto a second glass slide. Removal of the new membrane exposed the oxic surface for ablation.

Approximately 200 ng of digested peptides resuspensions were separated through nano-LC (1260 Infinity LC System, Agilent, Santa Clara, CA) with a C18 column (Agilent, Zorbax 300SB, and column dimensions of 150 mm length × 75 μm internal diameter, with 3.5 μm pore size particles) and a 90-min increasing organic gradient from 7% to 95%. The aqueous

phase was 0.1% formic acid in water and the organic phase was 0.1% formic acid and 80% acetonitrile in water, leading to a total run time of 120 min at a flow rate of 300 nL/min.

The chromatographically separated peptides were then electrosprayed at 1.90 kV spray voltage and 275˚C capillary temperature. Peptide ions were analyzed in a high mass resolution tandem mass spectrometer (Orbitrap Velos Pro, Thermo Fisher Scientific, Waltham, MA) with automatic gain control set at $10^6$ ions and an injection time between 1 and 200 ms. Full-scan mass spectra were collected in data-dependent acquisition mode from m/z 400–2000 at 30,000 mass resolution. Ten precursor ions were selected from each full mass spectrum for MS/MS analyses in high energy collision-induced dissociation mode using 30% energy for fragmentation.

The MaxQuant computational platform [44] was deployed for protein identification using the UniProt Knowledgebase [45], which incorporates the data curated by Pseudomonas.com [46]. KEGG identifications were obtained using Pseudomonas.com if the PseudoCap identification was not found on UniProt. The Perseus computational platform [47] was used for statistical analysis via label free quantification (LFQ), which employs peptide ion intensities to represent protein abundances. Analysis of proteomic data generally followed the previously described procedure with only slight modifications [8], as summarized below. Mass spectra data (.raw) files were directly uploaded and processed using a FASTA file of 3696 reviewed *P. aeruginosa* proteins (taxonomy identification 287) downloaded from the UniProt Knowledgebase database source and used for protein identification via MaxQuant. The main search parameters for MaxQuant identification were set at peptide tolerance of 4.5 ppm, MS/MS match tolerance of 20 ppm, MS/MS *de novo* tolerance of 10 ppm, minimum peptide length of 7, false discovery rate of 0.01, carbamidomethyl fixed modification, oxidation and acetylation (N-terminus) variable modifications, and enabled contaminant search. Perseus performs label free quantitation starting with ion intensities from "RAW" files generated by the Orbitrap mass spectrometer then converting these into a semi-quantitative measure of protein abundance [47]. An unpaired Students' t-test was used in Perseus, with permutation based false discovery rate of 0.05 in order to reduce identifications of false positives. The final set of identified peptides was further reduced in a spreadsheet by application of two additional selection criteria: ≥10% sequence coverage and at least two unique peptides.

The ablated material from three biological replicates of transwell membrane biofilms were analyzed separately by LC-MS/MS and the output averaged to represent the axially-resolved proteome. The proteins from oxic vs. anoxic regions were compared to identify statistically significant (q-value ≤ 0.05, and a corresponding adjusted p-value) high and low abundance proteins with at least a twofold change. No statistics are provided for the single analysis for each region by LC-MS/MS of polycarbonate membrane grown biofilms. The resultant radially-resolved proteome is only preliminary, but is reported here nonetheless in case it might be of use for future studies.

## Results and discussion

### Verification of proteins ablated from polycarbonate membrane biofilms

Preliminary studies required identification of the presence of ablated protein in the material laser ablated from polycarbonate membrane-grown biofilms. This identification was performed with SDS-PAGE on ablated cell lysates that were each pooled from six distinct biofilms of total areas of 6, 12, 18 and 24 mm$^2$. These areas correspond the perimeters of 1–4 mm$^2$ squares ablated from the same position on distinct biofilms, then multiplied by six to account for the pooling of sample collected from six separate biofilms. The images of the protein bands on a polyacrylamide gel from each lane from the pooled samples are shown in Fig 4A and

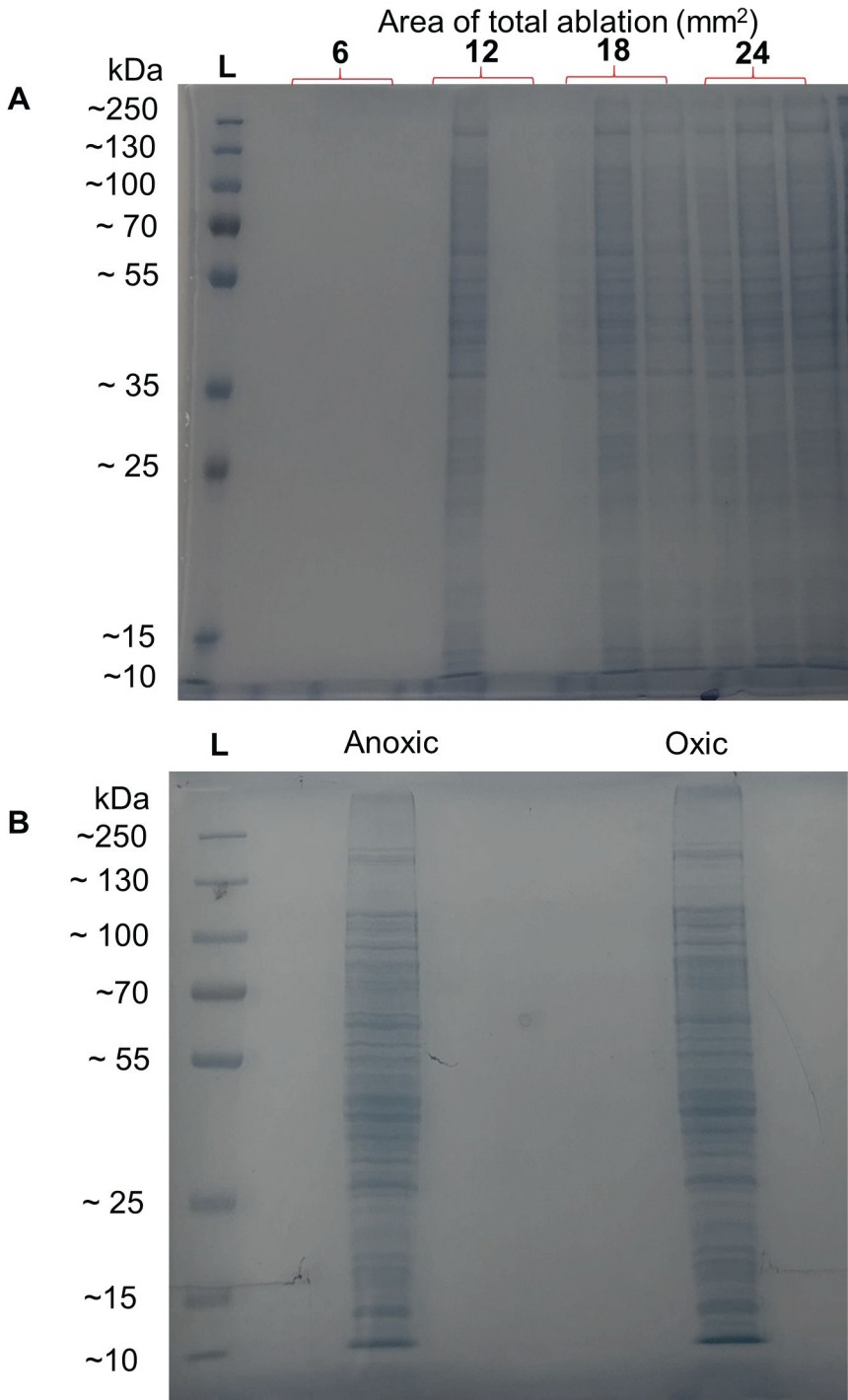

**Fig 4. Gels from SDS-PAGE of cell lysates from laser ablated biofilms.** A) Preliminary identification of the presence of proteins in cell lysates from biofilms grown on polycarbonate membranes, collected by laser ablation from 6–24 mm² total biofilm area. B) Confirmation of proteins in cell lysates collected by laser ablated from anoxic (left) and oxic (right) 24 mm² area regions of biofilms grown on transwell membranes. Protein bands of gel lanes were subsequently excised for in-gel digestion and bottom-up proteomic analysis.

confirm the ability of LAST to sample small areas of *P. aeruginosa* biofilms. Clearly distinguishable bands were observed on lanes resulting from at least 12 mm$^2$ of total biofilm area. These 12 mm$^2$ lanes subsequently yielded a considerable number of protein identifications via LC-MS/MS, as discussed below. However, 24 mm$^2$ of ablated material was designated as the minimum required for the oxic vs. anoxic analyses by LC-MS/MS to improve protein coverage.

### Preliminary radially-resolved proteome from *P. aeruginosa* biofilms

The biofilms evaluated for their radially-resolved proteome were grown for three days on polycarbonate membranes placed on CSP-containing agar plates. This biofilm cultivation method establishes opposing concentration gradients of oxygen and media components similar to chronic wounds and was methodologically convenient for generating the multiple biofilms necessary for development of the methods described herein [48–50]. Furthermore, growth on polycarbonate membranes is analogous to biofilm growth on membranes in tissue culture wells [8, 14].

Radially-resolved proteomic analyses were performed by collecting ablated material from two radially distinct regions of three day old polycarbonate-grown biofilms: a central biofilm region which corresponded to older cells (brown square on Fig 5A) and an outer region of the biofilm which corresponded to younger cells (green square on Fig 5A) as biofilms grew radially outward from the central inoculation site. Fig 5A shows schematically that ablation was actually performed in the shape of concentric squares, a pattern that was readily available to the motion stage while minimizing the age distribution of cells sampled. Fig 5B shows an optical image of post-ablation markings on an actual biofilm.

LC-MS/MS analysis of the ablated material from these preliminary samples resulted in identification of ~400 proteins, but only ~200 of them satisfied the more stringent criteria of at least three unique peptides and ≥10% sequence coverage. Analysis of single samples pooled from six biofilms found 86 proteins unique to the newer cell region (24 mm$^2$ total area) and fewer proteins unique to the older cell region, as expected given that the latter's 12 mm$^2$ area was only half that of the former. The analyses carried out on radial ablations from biofilms on polycarbonate membranes were performed mainly to support method validation, but these preliminary protein identities are documented in S1 Table in S1 File. All further work described below focused on the younger cell region of the transwell membrane-grown biofilms.

### Oxic and anoxic regions of *P. aeruginosa* biofilms display major differences in proteome

Fig 4B shows SDS-PAGE of cell lysates from 24 mm$^2$ area regions that were laser ablated from anoxic and oxic regions of biofilms grown on transwell membranes. While visual examination of the gel bands of Fig 4B showed no apparent differences between laser ablation of anoxic vs. oxic regions of transwell-grown biofilms, bottom-up proteomics by LC-MS/MS displayed very clear differences between the two regions. 24 mm$^2$ square perimeters were laser ablated from three anoxic or oxic regions of six transwell-grown biofilms grown (three days, all from the same culture). Proteins from each sample was separated via SDS-PAGE, then the proteins were digested on the gel and all bands of the lane were extracted and the resultant peptides injected into the LC-MS/MS. All proteins identified by this procedure are tabulated in S2 Table in S1 File along with their three or four-letter codes, full protein names, the number of unique peptides identified, the sequence coverage, the differential abundance for anoxic vs. oxic regions, and statistical significance.

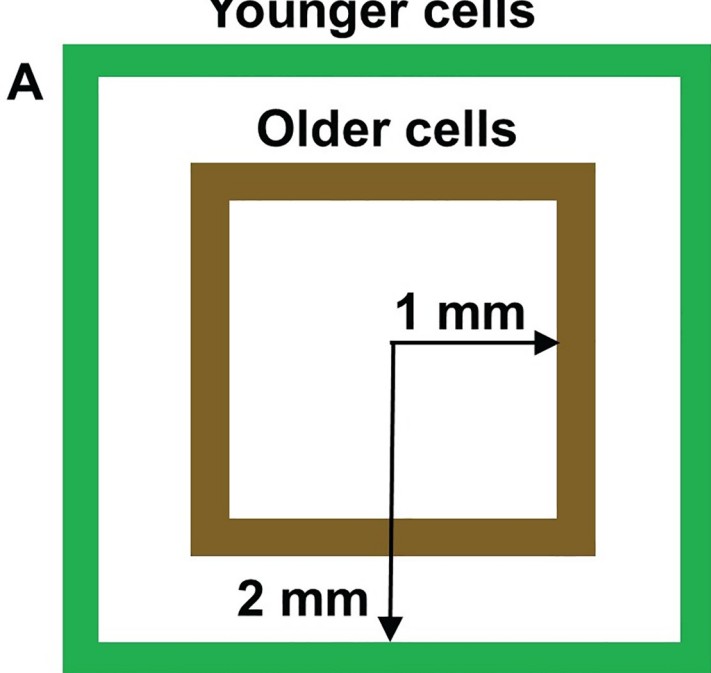

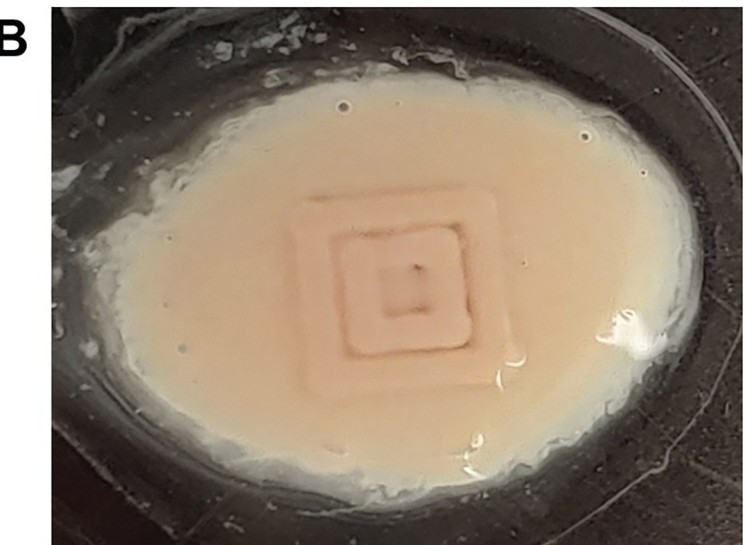

**Fig 5. Laser ablation of biofilms in square perimeter patterns.** A) Schematic of concentric ablation squares that are either 1 mm (older cell region, in brown) or 2 mm (younger cell region, in green) from the center of the biofilm, ablation of polycarbonate membrane biofilms. B) Optical image of post-ablation marks on polycarbonate biofilm.

LAST combined with LC-MS/MS revealed proteins with statistically significant different abundances between anxoic and oxic regions of these transwell biofilms. The volcano plot in Fig 6 includes 250 proteins identified that satisfied the stringency conditions of at least two peptides and $\geq$10% sequence coverage, out of a total of 430 proteins identified. An oxygen-selective electrode distinguished the oxic from anoxic regions in biofilms of this same strain grown in the same medium and cultivation system (albeit for a shorter growth period) [8], but

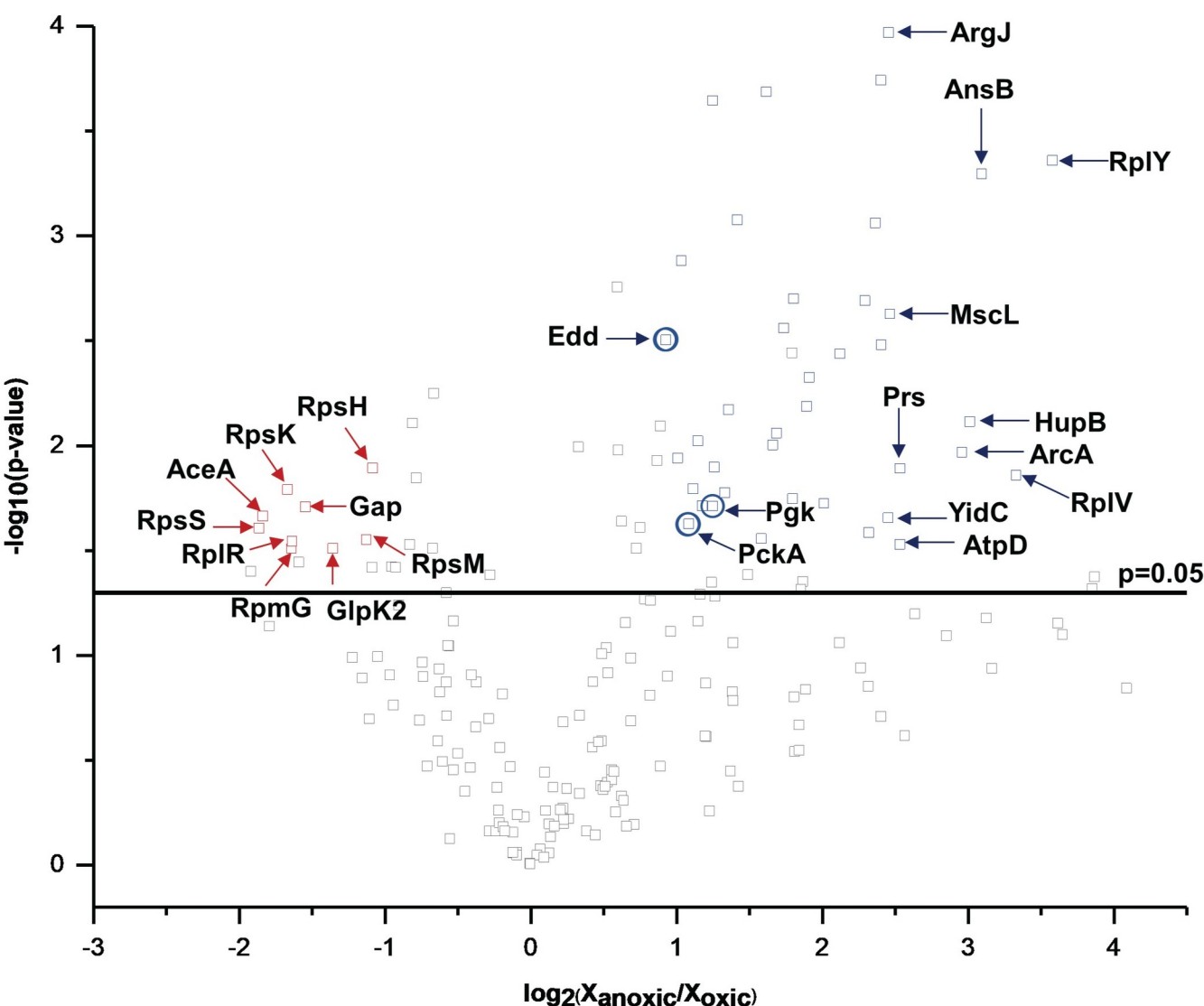

**Fig 6. Volcano plot of statistically significant, differentially expressed proteins from transwell-membrane biofilms.** Differential protein abundances for anoxic vs. oxic regions of *P. aeruginosa* three day grown transwell biofilms. Proteins with statistically significant ($-\log_{10}$(p-value) > 1.3) changes in abundance are highlighted above horizontal black line. Of those, proteins with $|\log_2(X_{anoxic}/X_{oxic})| \geq 1$, where $X_{(an)oxic}$ is protein abundance, are deemed as significantly higher or lower abundances and are annotated as blue or red points, respectively. The ten proteins that displayed the greatest fold changes in abundance are labelled with their three or four-letter protein codes (S2 Table in S1 File). Glucose catabolizing proteins that display significant changes in abundance are also circled (Edd, Pgk, and PckA).

the proteins observed therein were averaged across the entire biofilm. The results presented here show that the bottom, anoxic region of the biofilm displayed the majority of increases in protein abundance. The bottom, anoxic portion of the *P. aeruginosa* biofilm had 39 higher abundance proteins relative to the top/oxic of the biofilm, while the oxic region had only nine higher abundance proteins relative to the bottom/anoxic region. The observed bottom vs. top differences in the proteins support the argument that LAST distinguishes anoxic from oxic regions. However, these experiments cannot rule out the possibility that cells from the two regions are at least partially mixed either in the native biofilms and/or as a result of the ablation process.

Previous analyses of vertical differences of *P. aeruginosa* biofilms indicated that essentially all transcripts were higher in the upper portion of the biofilm [6, 14]. The observation of the opposite trend here is somewhat unexpected, as *P. aeruginosa* relies largely on respiration for energy generation and the CSP medium did not contain alternative electron acceptors like nitrate, constraining respiration in the lower regions of the biofilm. Alternatively, the higher abundance of proteins in the anoxic region might derive from its proximity to the growth media [10] which might drive catabolism more than oxygen abundance.

Two to threefold higher abundances of ribosomal proteins (RpsS, RpsK, RpmG, RpsM, RpsH and RplR) were observed in the anoxic zone, consistent with the widely held hypothesis of minimal growth in anoxic zones. However, higher abundances of many metabolic proteins suggested the cells were generating cellular energy and acclimating to stressful chemical environments [51]. The anoxic region of the biofilm displayed roughly twofold higher abundances of glucose catabolizing proteins (Edd, PckA, and Pgk) as compared to the oxic region: these are also marked on Fig 6. *P. aeruginosa* catabolizes glucose using the Entner-Doudoroff glycolysis pathway (Edd and Eda) and the data suggest cells in the anoxic regions were catabolizing the sugar and likely producing byproducts like acetate. Cells in the oxic region of the biofilm had an almost fourfold higher abundance of the key acetate metabolic protein, AceA, from the glyoxylate shunt, suggesting either acetate from the anoxic region may be metabolized in the oxic zone or that there is mixing of cells from the two regions. A higher abundance of AceA implies that the oxic zone might play a dominant role in the carbon catabolite strategy known as reverse diauxie which prefers organic acids to glucose and which was observed previously in *P. aeruginosa* biofilms [8].

The cells in the anoxic region also displayed a fivefold higher abundances of proteins associated with adenosine triphosphate (ATP) synthase proteins (AtpD and AtpG). The data suggest there may be more than substrate level phosphorylation occurring as operating an oxidative tricarboxylic acid (TCA) cycle and ATP synthase requires external electron acceptors [52, 53]. The soluble transhydrogenase protein (SthA) also displayed a twofold higher abundance in the anoxic region potentially linking the conversion of anabolic and catabolic electrons. The exchange of redox active metabolites could enable respiration at a distance using small molecules like phenazines, although proteins associated with these processes were not identified [54, 55].

Cells in the anoxic portion of the biofilm displayed higher abundances of proteins from arginine and polyamine metabolism. CSP medium contained arginine and it would be preferentially available to cells on the bottom of the biofilm. Arginine is an important amino acid for *P. aeruginosa* acclimation to biofilm conditions and can be used for substrate level energy generation, pH modulation via deamination (ArcA) and decarboxylase reactions (SpeA) [56, 57]. Arginine also serves as a precursor for synthesis of putrescine and spermidine which stabilize cellular membranes to stress [56, 57]. Multiple arginine proteins were associated with arginine succinyltransferase, the arginine decarboxylase/agmatine deiminase pathway, and arginine biosynthesis including almost eightfold higher abundances of AnsB and ArcA proteins in the anoxic region; fivefold higher abundances of ArgC, ArgG, ArgJ, and SpuD; fourfold higher abundances of AruC and GdhB; and twofold higher abundances of ArgS and ArcB. Additionally, threefold higher abundance of cytosolic aminopeptidase protein (PepA) in the anoxic regions which could be degrading cellular proteins to repurpose the amino acids for new proteins, stress responses like pH mediation, and/or cellular energy synthesis.

A large number of higher abundance proteins in the anoxic region could also support virulence and environmental stress mediation. Some of these proteins include large-conductance mechanosensitive ion channels (MscL, fivefold higher abundance in the anoxic region) a type of protein transport system which is sensitive to mechanical perturbations and changes in

osmotic pressure [58]. Multidrug resistance protein (MexA) displayed twofold higher abundance in the anoxic region consistent with previous transcript analyses of anoxic cultivation which indicate *P. aeruginosa* is more tolerant to antibiotics in hypoxic environments [59]. Membrane protein insertase (YidC) is a ubiquitous protein from the general secretory pathway and aids in the translocation of proteins across the cytoplasmic membrane. It functions together with other proteins to promote insertion of membrane proteins [60]. It is also responsible for modulating the expression of hypoxic genes and when present can aid in the assembly of ATP synthases (such as AtpD, AtpG, and AtpA that were identified in this study). YidC also led to roughly tenfold lower abundances of some ribosomal proteins (RplY and RplV) in the oxic region of the biofilm [61]. DNA-binding protein (HupB) was eightfold higher abundance in the anoxic region. HupB are histone-like small proteins—also known as heat unstable (HU) proteins—that prevent DNA damage caused by oxidative stress and enable adaptation to stress response [62]. HU proteins can also control the transcription of a gene required for anoxic respiration [63]. The chaperone protein ClpB was twofold higher abundance in the anoxic region and can rescue stress damaged proteins in an ATP consuming process [64]. The energy requirement to maintain protein function under environmental stress may partially explain the apparent substantial energy metabolism occurring in the anoxic region of the biofilm. Finally, the extracellular polymeric substance (EPS) producing protein, AlgC, was threefold higher abundance in the anoxic zone, possibly driving resource flux toward additional EPS synthesis [65].

## Conclusions

This study demonstrated several important advantages of laser ablation sample transfer for regio-specific determination of a biofilm proteome. Prior work by some of the authors required the harvest of entire *P. aeruginosa* bacterial biofilms of several cm$^2$ area to achieve an average characterization of phenotype via proteomics [8]. However, that prior work did not distinguish the changes in the proteome that occurred as oxygen abundance varied with depth [8]. LAST is shown here to distinguish protein response in the anoxic vs. oxic regions selected by the flipping of the biofilm to measure a phenotypic difference between the two regions. LAST also allows regio-specific sampling across the surface of biofilm by the use of an ~260 µm laser irradiation spot diameter.

There are other experimental advantages of the LAST strategy. Ablated material can be concentrated via vacuum centrifugation, then stored at -80°C to prevent deterioration until analysis can be conveniently performed. Sample processing for analysis is carried out off-line by standard LC-MS/MS-based bottom-up proteomics. Furthermore, the version of the LAST setup deployed here is on wheels, allowing it to be moved into a bio-safety cabinet when used with pathogenic bacteria. Furthermore, the setup is remotely operated, avoiding exposure of personnel to ablated pathogenic aerosols.

Rapid dehydration and flaking of the biofilm can occur when the laser beam directly impacts a surface in reflection geometry, leading to a disruption of the water-based laser ablation process [66]. This problem was addressed here by inverting the microscope glass slide on which the sample rests so that the laser radiation penetrated through the glass. Transmission geometry is also advantageous for heat diffusion [30], minimizing the potential for thermal degradation of the biofilm. Furthermore, the sticky and mucous-like texture of the biofilm can minimize the collection of ablated material via reflection geometry, while the transmission geometry assists collection of ablated material via spallation.

There are considerable advantages to LAST compared to direct MS imaging techniques such as MALDI [15, 17–19, 67] or DESI [21, 22], despite LAST's lower lateral resolution. The

delocalization introduced by the matrix [68] or the solvent spray [69] and the less efficient desorption of large biomolecules [70] in direct MS imaging methods are eliminated using LAST. Direct MS imaging techniques have been combined with advanced instrumentation such as ion mobility and Fourier transform ion cyclotron resonance MS [15, 71, 72] to simultaneously achieve high protein coverage and lateral resolution, but such strategies are still not widely available. By contrast, a general LC-MS/MS workflow combined with LAST provides a significantly high number of protein identities using relatively low cost and widely accessible instrumentation, albeit at a relatively low lateral resolution.

Only a small region of the biofilm was needed for LAST compared to previous work that required several entire biofilms for proteomic analyses [8]. About 430 proteins were identified here by laser ablation of the surface region of a 24 mm$^2$ area of biofilm. This is comparable to the ~600 proteins identified from analysis of the cell lysate from three entire biofilms (of several cm$^2$ total area, where the entire thickness of the biofilm was required) [8]. Loosening of the protein identification criteria (i.e., the 10% sequence matching) would have further increased the number of protein identifications.

There has been long term interest in identifying spatial differences during biofilm development including clues that identify chemical heterogeneity therein [9]. The novel results presented here develop the radial and axial resolved proteome in important biofilm-forming bacteria, *P. aeruginosa*, which provides key insight to metabolic activity in the slow growing anoxic zone that was not apparent from previous transcript studies. An extension of this work to multi-species biofilms will provide insights to synchronize between the omics data to highlight the synergy in biofilms that are ubiquitous in the environment.

## Supporting information

**S1 File. This is the main Supporting information file.** S1 Fig depicts the strategy for exposing anaerobic surface of a biofilm. S1 Table lists the *P. aeruginosa* proteins identified from three-day biofilms grown on polycarbonate membranes. S2 Table lists the proteins identified from axial ablations on transwell membrane insert grown biofilms.
(PDF)

**S1 Raw images. Raw gel images.** These are the unprocessed gel images from SDS-PAGE.
(PDF)

**S1 Fig.**
(TIF)

## Author Contributions

**Conceptualization:** Fabrizio Donnarumma, Kermit K. Murray, Ross P. Carlson, Luke Hanley.

**Data curation:** Aruni Chathurya Pulukkody.

**Formal analysis:** Aruni Chathurya Pulukkody, Luke Hanley.

**Funding acquisition:** Ross P. Carlson, Luke Hanley.

**Investigation:** Aruni Chathurya Pulukkody.

**Methodology:** Aruni Chathurya Pulukkody, Yeni P. Yung, Fabrizio Donnarumma, Kermit K. Murray, Luke Hanley.

**Project administration:** Ross P. Carlson, Luke Hanley.

**Resources:** Luke Hanley.

**Supervision:** Ross P. Carlson, Luke Hanley.

**Validation:** Aruni Chathurya Pulukkody, Fabrizio Donnarumma.

**Visualization:** Aruni Chathurya Pulukkody.

**Writing – original draft:** Aruni Chathurya Pulukkody, Fabrizio Donnarumma, Luke Hanley.

**Writing – review & editing:** Aruni Chathurya Pulukkody, Fabrizio Donnarumma, Kermit K. Murray, Ross P. Carlson, Luke Hanley.

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
