## [Decision Letter · Decision Letter 0]

30 Apr 2021

PONE-D-21-11668

Spatially resolved analysis of Pseudomonas aeruginosa biofilm proteomes measured by laser ablation sample transfer

PLOS ONE

Dear Dr. Hanley,

Thank you for submitting your manuscript to PLOS ONE. After careful consideration, we feel that it has merit but does not fully meet PLOS ONE’s publication criteria as it currently stands. Therefore, we invite you to submit a revised version of the manuscript that addresses the points raised during the review process.

Please pay particular attention to the fact that PLoS One requires that data are made publicly available, and we are lucky enough in the proteomic field to have very good public repositories such as PRIDE to perform this duty

We look forward to receiving your revised manuscript.

Kind regards,

Thierry Rabilloud

Academic Editor

PLOS ONE

Journal Requirements:

2. Please include details of the sources of the bacterial cell lines in your methods section.

Reviewers' comments:

Reviewer's Responses to Questions

**Comments to the Author**

1. Is the manuscript technically sound, and do the data support the conclusions?

Reviewer #1: Yes

Reviewer #2: Yes

2. Has the statistical analysis been performed appropriately and rigorously? 

Reviewer #1: Yes

Reviewer #2: Yes

3. Have the authors made all data underlying the findings in their manuscript fully available?

Reviewer #1: No

Reviewer #2: No

4. Is the manuscript presented in an intelligible fashion and written in standard English?

Reviewer #1: Yes

Reviewer #2: Yes

5. Review Comments to the Author

Reviewer #1: General remarks

This work presents a spatially resolved characterization of biofilm proteomes from P. aeruginosa. The authors present comprehensive results based on site-specific sampling of biofilms and proteome analysis. The methodological challenges and limitations are honestly discussed and the functions of differently abundant proteins in various cellular processes are described in detail.

The experiments performed in this study are well performed and quality of the data-set is ensured by valid filter criteria.

However, currently raw-data can be accessed which makes it impossible to judge on the quality of the provided material. Small improvement in grammar and style and the discussion might enhance the quality of the manuscript.

Therefore, I suggest a minor revision before accepting this manuscript for publication in PlosOne.

Detailed remarks

• The available data in the MassIVe repository is private. Please provide the reviewer login-details to allow inspection of the provided files.

• Please add a reference to the used strain isolate in the experimental section.

• Did the authors use ion intensities (page 7, last sentence) or LFW (page 8, first paragraph) to determine protein abundance, or both?

• Please provide the number of entries in the proteome database actually searched in the experimental section.

• How was the FDR calculated?

• From figure 2 a it seems as there have been 2 or even 3 samples per area. Are those replicates? If so, which kind of replicate?

• Page 11: “This was in stark contrast to previous analyses of vertical differences in transcriptome of P. aeruginosa biofilms which indicated essentially all transcripts were higher in the upper portion of the biofilm.”. Please discuss on this issue.

• Please add glucose catabolizing proteins (page 12, first paragraph) to figure 4.

• Proteins involved in glucose catabolism and substrate level phosphorylation are higher abundant in anoxic regions. This seems to be unexpected to me. Please discuss.

• Please write the organism’s names completely if you are mentioning them the first times and use abbreviation of the genus only later (e.g. end of page 3).

• Please add reference to figure 2 b to paragraph A of the results section.

• The terms “up-/downregulated” match gene expression but do not describe the changes in protein abundance, which have been determined here. No direct information on “regulation” can be obtained from a proteomic data set. I suggest to use terms which reflect this issue, e.g. enriched, depleted, more/less abundant.

• Page2: “…metabolic activity has been measured using reporter proteins like green fluorescent protein and transcriptome”. Please check style and be consistent in the level of method description you are using.

• Page 6: “Next, the glass slide then placed in a secondary container and…”. There is a verb missing.

• Page 6: “The bands on each well represents the separation of proteins from the ablated material.” Sorry but I don’t get the meaning of this sentence.

• Please check style: “Such nosocomial infections can transition from acute to chronic stages..” (page2), “This study reveals active metabolism in the anoxic region of the biofilm with respect to the oxic region in P. aeruginosa” (abstract)

Reviewer #2: This paper focuses on the use of the interesting LAST method coupled with LC-MS/MS to gain information on the proteins of Pseudomonas aeruginosa biofilms. The manuscript is clear and well written.

Figure S1 should be the first figure in the article. It provides an overview of the system used to perform these experiments.

It does not appear to me that the authors have deposited the data on a public repository. I highly advise the authors to deposit these data.

The authors use a database downloaded from Uniprot (page 8). How many proteins are in this database? Why not use the dedicated website www.pseudomonas.com ? Has the strain used been sequenced?

Page 8, it is indicated "acetylation". On which amino acid (K or N-terminal) ? Please specify.

For quantification, page 8, it is indicated a factor of at least 1.5 for protein selection. Generally, a factor equal to 2 is used. Why this choice of 1.5? I do not understand the choice of the factor of 1.3 in the legend of figure 4. Why not use 1.5? Please correct this figure by taking into account proteins with factors higher than 1.5 (or 2 depending on the answer to my previous question).

I find part C (page 10) interesting but it lacks relevant information. The authors describe groups of more (or less) abundant proteins between conditions. But what are the differences between the different studied regions of the biofilm? Are there more specific mechanisms? Are there proteins more involved in biofilm formation? I think highlighting this information more would improve the manuscript.

6. PLOS authors have the option to publish the peer review history of their article (what does this mean?). If published, this will include your full peer review and any attached files.

Reviewer #1: No

Reviewer #2: No

---

## [Decision Letter · Decision Letter 1]

23 Jun 2021

Spatially resolved analysis of Pseudomonas aeruginosa biofilm proteomes measured by laser ablation sample transfer

PONE-D-21-11668R1

Dear Dr. Hanley,

We’re pleased to inform you that your manuscript has been judged scientifically suitable for publication and will be formally accepted for publication once it meets all outstanding technical requirements.

Kind regards,

Thierry Rabilloud

Academic Editor

PLOS ONE

Additional Editor Comments (optional):

Reviewers' comments:

Reviewer's Responses to Questions

**Comments to the Author**

1. If the authors have adequately addressed your comments raised in a previous round of review and you feel that this manuscript is now acceptable for publication, you may indicate that here to bypass the “Comments to the Author” section, enter your conflict of interest statement in the “Confidential to Editor” section, and submit your "Accept" recommendation.

Reviewer #1: All comments have been addressed

2. Is the manuscript technically sound, and do the data support the conclusions?

Reviewer #1: Yes

3. Has the statistical analysis been performed appropriately and rigorously? 

Reviewer #1: (No Response)

4. Have the authors made all data underlying the findings in their manuscript fully available?

Reviewer #1: Yes

5. Is the manuscript presented in an intelligible fashion and written in standard English?

Reviewer #1: Yes

6. Review Comments to the Author

Reviewer #1: (No Response)

7. PLOS authors have the option to publish the peer review history of their article (what does this mean?). If published, this will include your full peer review and any attached files.

Reviewer #1: No

---

## [Editor Report · Acceptance letter]

13 Jul 2021

PONE-D-21-11668R1 

Spatially resolved analysis of *Pseudomonas aeruginosa* biofilm proteomes measured by laser ablation sample transfer 

Dear Dr. Hanley:

I'm pleased to inform you that your manuscript has been deemed suitable for publication in PLOS ONE. Congratulations! Your manuscript is now with our production department. 

Kind regards, 

on behalf of

Dr. Thierry Rabilloud 

Academic Editor

PLOS ONE